# Crystal structures of Ryanodine Receptor reveal dantrolene and azumolene interactions guiding inhibitor development

Hadiatullah Hadiatullah [1,9], Lianyun Lin[1,9], Zhiyan Wang[2,9], Rajamanikandan Sundarraj[1], Qing Wang[1], Xinru Lai[1], Nagomi Kurebayashi [3], Takuya Kobayashi [3], Toshiko Yamazawa[4], Yu Seby Chen [5], Wenlan Wang[1], Hongxia Zhao[1], Yiqing Yin[6], Takashi Murayama [3], Filip Van Petegem [5] & Zhiguang Yuchi [1,2,7,8] ✉

The ryanodine receptor (RyR) is a critical drug target, yet dantrolene (DAN) remains the only FDA-approved inhibitor, limited by hepatotoxicity and unsuitable for chronic use. To guide improved inhibitor development, we determine high-resolution crystal structures of the RyR Repeat12 (R12) domain bound to DAN, its analog azumolene (AZU), and adenine nucleotides (AMP-PCP or ADP). DAN/AZU and nucleotides bind cooperatively to a pseudosymmetric cleft, with key interactions involving Trp880 and Trp994. Binding induces a clamshell-like closure of the R12 domain. Isothermal titration calorimetry (ITC) reveals higher affinity in the presence of nucleotides and lower affinity for RyR2 due to nearby substitutions. Structural comparison with cryo-EM data suggests that DAN/AZU binding allosterically influences RyR gating and functional regulation. Structure-based screening identifies a potent compound targeting the same site but with a distinct binding mode. Our findings highlight the power of domain-focused crystallography in guiding RyR inhibitor discovery and overcoming cryo-EM resolution limitations.

Ryanodine receptors (RyRs) are vital calcium channels located in the endoplasmic (ER) and sarcoplasmic reticulum (SR) membranes that play a key role in mediating excitation-contraction (EC) coupling in muscle cells[1,2]. Among the three isoforms, RyR1 is primarily expressed in skeletal muscle, RyR2 predominates in cardiac muscle, and RyR3 has a more ubiquitous expression pattern. However, all three isoforms are found in multiple tissues[3–5]. Mutations in RyR1 are linked to severe myopathies, including malignant hyperthermia (MH) and central core disease (CCD), both of which are associated with dysfunctional calcium homeostasis[6]. These conditions often result from gain-of-function mutations that cause chronic calcium leakage or aberrant activation. In MH, this dysregulation is triggered by volatile

[1]State Key Laboratory of Synthetic Biology; Frontiers Science Center for Synthetic Biology; Tianjin Key Laboratory for Modern Drug Delivery & High-Efficiency; School of Pharmaceutical Science and Technology, Faculty of Medicine, Tianjin University, Tianjin, China. [2]Department of Molecular Pharmacology, Tianjin Medical University Cancer Institute & Hospital; National Clinical Research Center for Cancer; Key Laboratory of Cancer Prevention and Therapy; Tianjin's Clinical Research Center for Cancer, Tianjin, China. [3]Department of Pharmacology, Juntendo University School of Medicine, Tokyo, Japan. [4]Core Research Facilities, The Jikei University School of Medicine, Tokyo, Japan. [5]Department of Biochemistry and Molecular Biology, Life Sciences Institute, University of British Columbia, Vancouver, BC, Canada. [6]Department of Anesthesiology, Tianjin Medical University Cancer Institute & Hospital; National Clinical Research Center for Cancer; Key Laboratory of Cancer Prevention and Therapy; Tianjin's Clinical Research Center for Cancer, Tianjin, China. [7]Haihe Laboratory of Sustainable Chemical Transformations, Tianjin, China. [8]Guangdong Laboratory for Lingnan Modern Agriculture (Shenzhen Branch), Agricultural Genomics Institute at Shenzhen, Chinese Academy of Agricultural Sciences, Shenzhen, Guangdong, China. [9]These authors contributed equally: Hadiatullah Hadiatullah, Lianyun Lin, Zhiyan Wang. ✉e-mail: yuchi@tju.edu.cn

anesthetics[7], whereas in CCD, persistent calcium leakage depletes SR stores, leading to muscle weakness[8]. RyR1 thus represents a compelling therapeutic target for addressing these disorders.

Dantrolene (DAN), a hydantoin derivative, is the primary treatment for MH, functioning as an inhibitor of RyR1-mediated calcium release[9,10]. While its direct binding to RyR1 has been demonstrated through [³H]azidodantrolene photolabeling experiments[11], the precise binding site and molecular mechanism of action remained incompletely understood until recently. The isoform specificity of DAN is controversial. It has been shown to reduce [³H]ryanodine binding to RyR1 and RyR3 but not RyR2, with its efficacy modulated by ATP, $Mg^{2+}$, and calmodulin (CaM)[12–14]. However, other studies suggest that DAN may inhibit RyR2 under specific conditions, such as in animal models of heart failure or arrhythmias, by modulating interdomain interactions[15–18]. Despite its clinical utility, DAN has two major limitations: hepatotoxicity, which restricts its long-term use for chronic conditions like CCD and sarcopenia[19,20], and poor solubility, which limits its therapeutic potential despite reformulations aimed at improving reconstitution rates[21]. Azumolene (AZU), a water-soluble analog of DAN, offers a promising alternative with similar potency and approximately 30-fold greater solubility; however, further structural characterization is needed to fully elucidate its mechanism of action[22,23].

Recent cryo-electron microscopy (cryo-EM) studies have revealed the modular architecture of RyRs, which are composed of over 20 domains, including four repeat domains arranged in tandem. Among these, Repeat12 (R12), located in the peripheral region of the cytosolic cap, plays a critical role in coupled gating, as well as in ligand binding and regulation[24]. It serves as a key binding site for ATP and inhibitory compounds such as Rycals, which are being explored for their therapeutic potential in skeletal and cardiac muscle disorders[25]. However, the intrinsic flexibility of R12, reflected in low-resolution cryo-EM density maps, poses challenges for accurately modeling ligand interactions. A recent cryo-EM structure of RyR1 in complex with DAN showed poor density for the molecule, precluding a detailed analysis of its binding mode[26].

In this work, we confirm that both DAN and AZU bind directly to the R12 domain of RyR1 and RyR3 with high affinity, as determined by isothermal titration calorimetry (ITC). Their binding is modulated by the presence of ATP or ADP, underscoring the role of nucleotide interactions in receptor regulation. We solve several high-resolution crystal structures of the RyR3 R12 domain, both in its apo form and in complex with DAN and AZU, in the presence of non-hydrolyzable ATP analog AMP-PCP or ADP, revealing their precise binding poses. The binding modes for DAN and AZU are at odds with a recently reported cryo-EM structure of DAN bound to RyR1[26], where the local resolution is poor. Our study underscores the usefulness of X-ray crystallography for detailed investigations of RyR-drug interactions. Complementary mutagenesis, ITC, and calcium imaging assays identify key residues involved in inhibitor binding and isoform-specific modulation. Leveraging our high-resolution complex structure, we perform virtual screening and identify a potent compound that effectively inhibits RyR1 by targeting its DAN-binding site, with its binding mode further validated by X-ray crystallography. These findings offer a structural framework for developing more effective RyR-targeting therapeutics to treat MH, CCD, and related myopathies.

## Results
### DAN and AZU bind to the R12 domain of RyR1 and RyR3 but not RyR2
To confirm the inhibitory effects of DAN and AZU (Fig. 1A) on full-length RyR, we employed a fluorescence-based assay that monitors ER $Ca^{2+}$ levels in HEK293 cells stably expressing RyR1 with the MH mutation R2163C. Both ligands effectively inhibited $Ca^{2+}$ leakage, with $IC_{50}$ values of 0.26 μM and 0.41 μM, respectively (Fig. 1B). To determine

whether the R12 domain serves as the direct binding site for DAN and AZU, we performed ITC experiments to evaluate their interactions with the R12 domain of the three RyR isoforms. Both DAN and AZU exhibited clear binding to the R12 domain of RyR1 with comparable apparent binding affinities. Their binding is driven by enthalpy, accompanied by an unfavorable entropic contribution (Fig. 1C, D, and Supplementary Table 1).

Strikingly, the presence of AMP-PCP or ADP significantly enhanced the binding affinities of DAN and AZU in a concentration-dependent manner. For DAN, the Kd decreased ~3,800-fold and ~16,000-fold in the presence of 500 μM and 5 mM AMP-PCP, respectively (Fig. 1C and Supplementary Table 1). A similar enhancement was observed for DAN in the presence of ADP, as well as for AZU binding under comparable conditions (Fig. 1C, D, and Supplementary Table 1). In the presence of 5 mM AMP-PCP, the Kd values for DAN and AZU were 22 nM and 62 nM, respectively (Fig. 1C and Supplementary Table 1), which closely align with the $IC_{50}$ values measured for full-length RyR1 (Fig. 1B)[27], reinforcing the idea that the R12 domain serves as the primary binding site for these ligands within RyR. Notably, all these interactions are driven by both favorable enthalpy and entropy, in contrast to the binding observed when DAN or AZU are alone (Fig. 1C, D, and Supplementary Table 1). It is worth mentioning that AMP-PCP or ADP alone did not exhibit detectable binding by ITC, likely due to their low affinity (Supplementary Fig. 1). This suggests a difference in their binding modes in the absence and presence of DAN/AZU, akin to the inverse scenario observed for DAN/AZU binding.

For the RyR2 R12 domain, neither DAN nor AZU exhibited detectable binding, even in the presence of AMP-PCP or ADP (Supplementary Fig. 2). This isoform-specific lack of binding is consistent with previous findings from [³H]ryanodine binding assays, calcium imaging, and lipid-bilayer experiments using full-length RyR2, which showed no inhibitory effect of DAN on RyR2[12,27–29]. However, some studies have suggested that DAN might inhibit RyR2, implicating a potential role for accessory proteins such as FKBP12.6 and CaM in DAN modulation[14,30]. Additionally, the effects of DAN on RyRs might be more pronounced under specific pathological conditions, such as MH or arrhythmia[15,16,31].

In contrast, DAN and AZU bound to the R12 domain of RyR3 with affinities and thermodynamic parameters comparable to those observed for RyR1, both in the absence and presence of AMP-PCP or ADP (Supplementary Fig. 3 and Supplementary Table 1). This is consistent with reports that RyR3 can also be inhibited by DAN[12], and that DAN has been explored for treating RyR3-related diseases, such as Alzheimer's disease[32–34]. Overall, our findings support that, in the isolated R12 domains of the three RyR isoforms, DAN and AZU preferentially bind RyR1 and RyR3.

### Crystal structures of RyR3 R12 in complex with DAN
The R12 domain is located at the corner of the cytosolic region of RyR, adjacent to the SPRY1 domain from the same subunit and the BSol domain from the neighboring subunit (Fig. 2A). Previous cryo-EM studies of all three RyR isoforms reported low local resolution at this region due to the high flexibility, likely limiting the precise modeling of the ligand binding[35–37]. Despite multiple attempts, we were unable to obtain diffraction-quality crystals for RyR1 R12 co-crystallized with DAN. In contrast, RyR3 R12 exhibited greater stability and crystallizability while maintaining similar binding properties to DAN and AZU as observed in RyR1. As a result, RyR3 R12 was selected for structural studies in this work (Supplementary Fig. 4). Throughout the text, residue numbers follow human RyR1 numbering, except in the RyR3 structural descriptions, where the corresponding RyR1 residues are indicated in brackets.

A cryo-EM structure of RyR3 was recently reported, but the local resolution in the R12 domain was particularly low (~6 Å)[35]. To address this, we started by solving a crystal structure of the individual domain at ~1.97 Å resolution (Supplementary Table 2), allowing for a high-

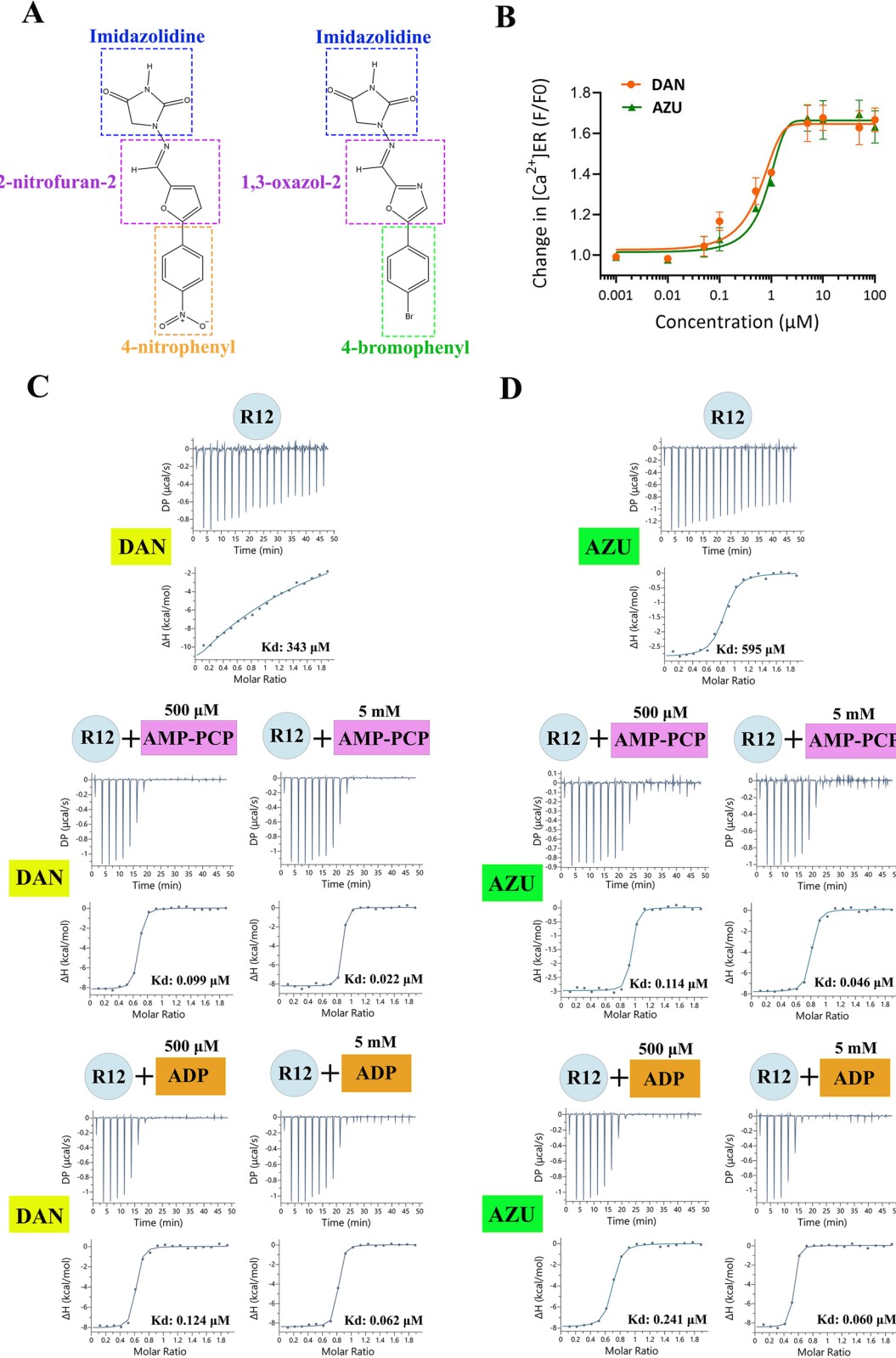

**Fig. 1 | DAN and AZU inhibit RyR1 through binding to the R12 domain.**
**A** Chemical structures of DAN and AZU. **B** Dose-dependent effects of DAN and AZU on ER luminal [Ca²⁺] changes in cell lines expressing RyR1 R2163C. The data are presented as the mean ± SD ($n = 3$ independent experiment). Source data are provided as a Source Data file. ITC binding isotherms illustrating the interaction of DAN (**C**) or AZU (**D**) with the R12 domain of RyR1 in the absence or presence of AMP-PCP or ADP. The affinity and thermodynamic parameters are listed in Supplementary Table 1.

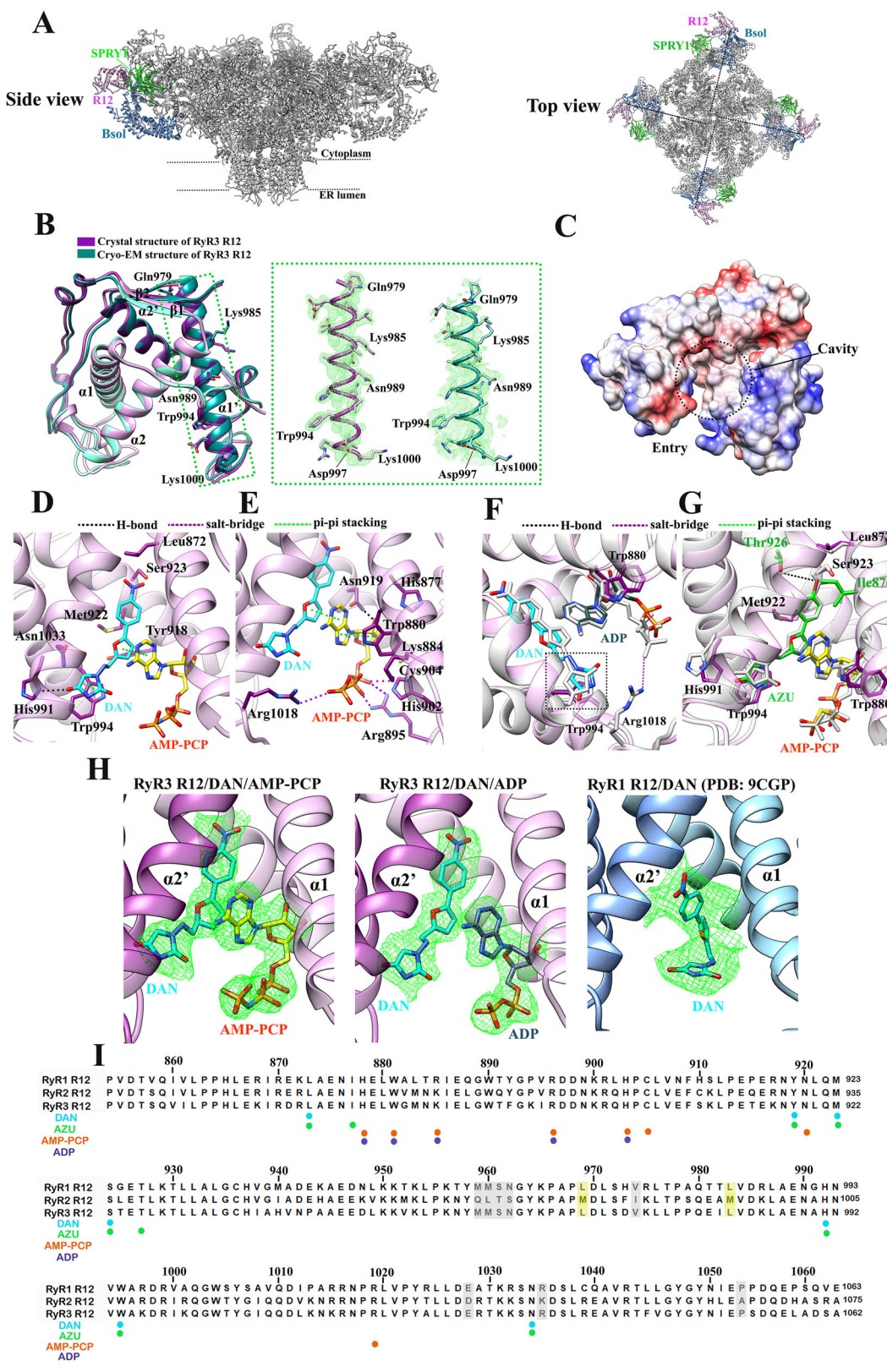

resolution view of this critical region. The crystal structure of RyR3 R12 closely resembles the cryo-EM-derived structure, with a root mean square deviation (RMSD) of ~0.3 Å for 174 Cα atoms (861–1034) (Fig. 2B). The RyR3 R12 domain exhibits a prominent horseshoe shape with pronounced twofold pseudosymmetry (Supplementary Fig. 5). Its two halves, R1 and R2, each contain two α-helices and a short C-terminal β-strand (Fig. 2B). A large cavity separates these halves, with

a hydrophobic core deep inside and a charged entry (Fig. 2C). Notably, several side chains adopt different conformations in the crystal structure compared to the cryo-EM model, including the key residues near the cavity such as Gln979 (Gln980 in RyR1), Lys985 (Arg986 in RyR1), Asn989 (Asn990 in RyR1), Trp994 (Trp995 in RyR1), Asp997 (Asp998 in RyR1), and Lys1000 (Ala1001 in RyR1). These differences highlight the advantage of X-ray crystallography in achieving atomic-

**Fig. 2 | Crystal structures of the R12 domain in complex with DAN and AZU.**
**A** Side and top views of RyRs showing the locations of the R12 domain, neighboring SPRY1 domain and Bsol domain. **B** Crystal structure of Apo RyR3-R12. Superposition of the crystal structure of RyR3-R12 (purple) and the cryo-EM structure of the R12 domain from full-length RyR3 (dark cyan) (PDB ID 9C1E) (left). Comparison of the electron density map (2Fo−Fc) for helix α1′ from the crystal structure and the corresponding cryo-EM density map, shown at σ levels of 1 and 5, respectively (right). **C** Electrostatic surface representation of the RyR3 R12 crystal structure, highlighting the cavity's location and hydrophobicity distribution. **D–H** Crystal structures of RyR3 R12 in complex with DAN/AZU and AMP-PCP/ADP. Residue numbers correspond to the RyR3 sequence. **D, E** Enlarged views from two angles illustrating the ligand interactions with R12 in the DAN (**D**) and AMP-PCP (**E**) binding regions. **F** Superposition of the crystal structures of RyR3 R12/DAN/ADP (plum) and

RyR3 R12/DAN/AMP-PCP (white). The imidazolidine moiety, which displays notable displacement, is indicated by a dashed box. **G** Superposition of the crystal structures of RyR3 R12/AZU/AMP-PCP (plum) and RyR3 R12/DAN/AMP-PCP (white). Additional contacts between AZU and the R12 residues Ile876 and Thr926 are highlighted by lines. All residue numbers are based on RyR3 sequence numbering. **H** The comparison of density maps of ligands from the structures of RyR3 R12/DAN/AMP-PCP (left), RyR3 R12/DAN/ADP (middle), and RyR1/DAN (right, PDB ID 9CGP), displayed at σ levels of 1, 1, and 5, respectively. **I** Sequence alignment of the R12 domains from three human RyR isoforms. Key coordinating residues in the binding sites of DAN (cyan), AZU (green), AMP-PCP (orange), and ADP (dark blue) are annotated at the bottom. Residue numbering above the alignment follows the human RyR1 sequence.

level accuracy, which is essential for elucidating ligand-binding mechanism and advancing structure-based drug design (Fig. 2B).

To elucidate the binding mechanism of DAN, we resolved the crystal structures of RyR3 R12 in complex with DAN in the presence of AMP-PCP and ADP at 2.79 Å and 2.84 Å resolutions, respectively (Supplementary Table 2). In the RyR3 R12/DAN/AMP-PCP structure, DAN binds to a site deep within the cavity, whereas AMP-PCP binds to a complementary site near the cavity entrance (Fig. 2D, E, and Supplementary Fig. 7A). The 4-nitrophenyl moiety of DAN inserts deeply into the cleft between helices α1 and α2 from R1, while its imidazolidine moiety, stabilized by a π-π stacking interaction with Trp994 (Trp995 in RyR1), points toward the pocket entrance and interacts with helices α1′ and α2′ from R2. This interaction is further supported by a π-π stacking interaction between the nitrofuran moiety of DAN and the adenine ring of AMP-PCP (Fig. 2D), explaining the increased affinity of DAN in the presence of AMP-PCP (Fig. 1C). On the opposite side of the adenine ring, a second π-π stacking interaction occurs between adenine and Trp880 (Trp881 in RyR1), effectively sandwiching the adenine ring between the nitrofuran moiety of DAN and Trp880 (Trp881 in RyR1). Additionally, three salt bridges between the triphosphate moiety of AMP-PCP and Lys884 (Arg885 in RyR1), Arg895 (Arg896 in RyR1) (from R1), and Arg1018 (Arg1019 in RyR1) (from R2), further stabilize AMP-PCP binding, reinforcing its interaction within the R12 domain (Fig. 2E).

We generated a homology model of the RyR1 R12 domain in complex with DAN and AMP-PCP, using our RyR3 complex structure as a template (Supplementary Fig. 6). Sequence alignment indicated that all ligand-contacting residues are conserved between RyR1 and RyR3, except for RyR1 Arg885, which is a lysine in RyR3 (Fig. 2I). Despite this substitution, both lysine and arginine can form a similar salt bridge with the triphosphate moiety of AMP-PCP, which is consistent with the comparable Kd and thermodynamic parameters observed in the ITC experiments. These findings validate the use of the RyR3 complex structure to study RyR1 modulation by DAN.

In the RyR3 R12/DAN/ADP structure (Fig. 2F and Supplementary Fig. 7B), the binding pose of DAN closely resembles that observed in the RyR3 R12/DAN/AMP-PCP structure. The primary difference lies in the imidazolidine moiety, which shifts toward the nucleotide. This shift is likely induced by the displacement of helix α2′, resulting from the loss of interaction between Arg1018 (Arg1019 in RyR1) and the terminal phosphate group of AMP-PCP (Fig. 2F). This difference in rearrangement may account for the slightly higher binding affinity of DAN in the presence of AMP-PCP compared to ADP (Fig. 1C, D, and Supplementary Table 1).

While preparing our manuscript, two cryo-EM structures of RyR1 bound to DAN were reported in both open and closed states[26]. Although the local resolutions for the R12 domain were reported as 3.8 Å (closed) and 3.2 Å (open), the ligand-binding region displayed low-quality density with indistinct blob-like features (Fig. 2H). Notably, DAN was modeled at a site near Trp881, which, in our high-resolution crystal structures, corresponds to the nucleotide-binding site. In their cryo-EM experiments, 2 mM ATP and 50 μM DAN were included in the sample buffer, raising the possibility that the observed density actually

corresponds to ATP rather than DAN. In contrast, the density maps from our crystal structures clearly distinguish DAN from nucleotides, with distinct shapes and features that precisely match their respective ligands. Furthermore, a comparison of our RyR3 R12/DAN/AMP-PCP and RyR3 R12/DAN/ADP structures revealed that the only difference in density maps is at the terminal phosphate group of AMP-PCP, validating that the density near Trp881 corresponds to nucleotides instead of DAN. Additionally, an unpublished RyR3 R12/AMP-PCP complex structure (PDB ID 6UHH) also confirms AMP-PCP binding at this site, further supporting that this region accommodates nucleotide rather than DAN, as suggested in the cryo-EM structures (Fig. 2H).

### Crystal structure of RyR3 R12 in complex with AZU

We demonstrated that AZU has $IC_{50}$ and Kd values comparable to those of DAN (Fig. 1B-D). To investigate whether AZU modulates RyR through a similar mechanism, we determined the crystal structure of RyR3 R12 in complex with AZU and AMP-PCP at a resolution of 3.00 Å (Supplementary Table 2). Both AZU and AMP-PCP simultaneously occupy the two binding sites, mirroring the arrangement observed for DAN. Trp880 (Trp881 in RyR1) and Trp994 (Trp995 in RyR1) serve as key anchoring points for AMP-PCP and AZU, respectively (Fig. 2G and Supplementary Fig. 7C).

However, differences in binding modes are observed. Compared with DAN, the bulky 4-bromophenyl moiety of AZU forms additional contacts with residues not engaged by DAN, including Ile876 (Ile877 in RyR1) and Thr926 (Thr927 in RyR1), positioning AZU deeper within the cleft (Fig. 2G). These unique interactions suggest potential for future modifications targeting this region to further increase the binding affinities of these ligands to the R12 domain.

### Validation of residues important for ligand binding and isoform specificity

To validate our structural model, we introduced mutations to two key tryptophan residues in the RyR1 R12 domain. ITC analysis revealed that the W995A mutation completely abolishes DAN/AZU binding (Fig. 3A and Supplementary Fig. 8A). Furthermore, even in the presence of AMP-PCP or ADP, DAN/AZU binding can no longer be detected, indicating that the adenine ring stacking interaction alone may be insufficient to sustain DAN/AZU binding in the absence of Trp995 (Fig. 3A, Supplementary Fig. 8A).

Conversely, the W881A mutation does not affect DAN binding in the absence of nucleotides, as its Kd value remains similar to that of the wild type (WT), suggesting that DAN alone binds the R12 domain without interacting with Trp881 (Fig. 3B, Supplementary Fig. 8B, and Supplementary Table 1). However, unexpectedly, W881A completely abolished DAN binding in the presence of the AMP-PCP or ADP (Fig. 3B, Supplementary Fig. 8B, and Supplementary Table 1). We speculate that the W881A mutation disrupts nucleotide binding at the Trp881 site, causing AMP-PCP/ADP to bind to an alternative lower-affinity site that overlaps with DAN-binding site near Trp995. This shift in nucleotide binding likely interferes with DAN binding, preventing its binding due

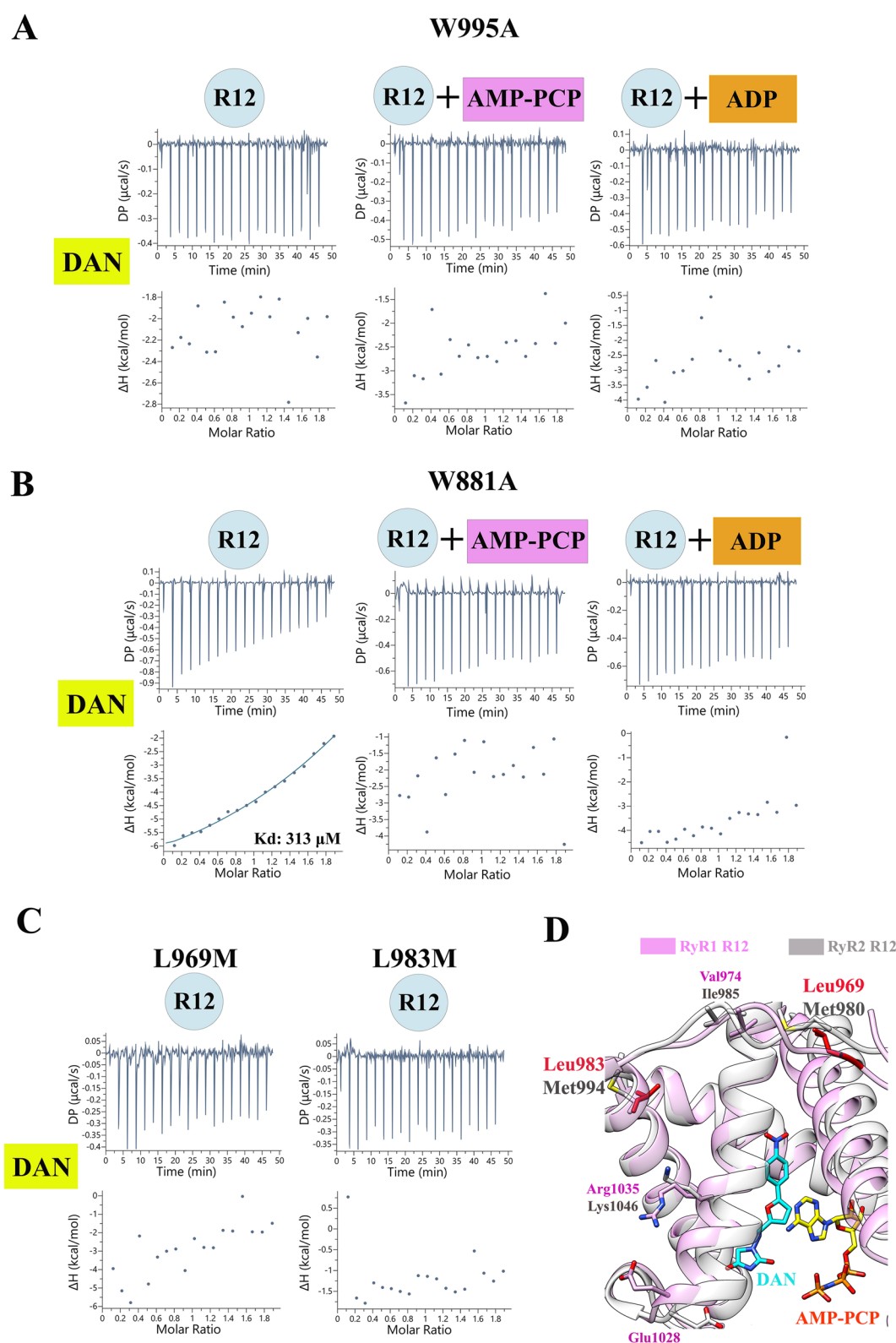

**Fig. 3 | Validation of key residues for ligand binding and isoform specificity.**
**A**–**C** ITC binding isotherms showing the interaction of DAN with the R12 domain of RyR1 harboring the W995A (**A**), W881A (**B**), L969M, and L983M (**C**) mutations. The affinity and thermodynamic parameters are listed in Supplementary Table 1.
**D** Superposition of the R12 domains from RyR1 (plum) and RyR2 (white). Homology models for both domains were generated using the RyR3 R12/DAN/AMP-PCP crystal structure as a template. Residues conserved between RyR1 and RyR3 but non-conserved in RyR2 are displayed, with key residues critical for isoform specificity, Leu983 and Leu969, highlighted in red.

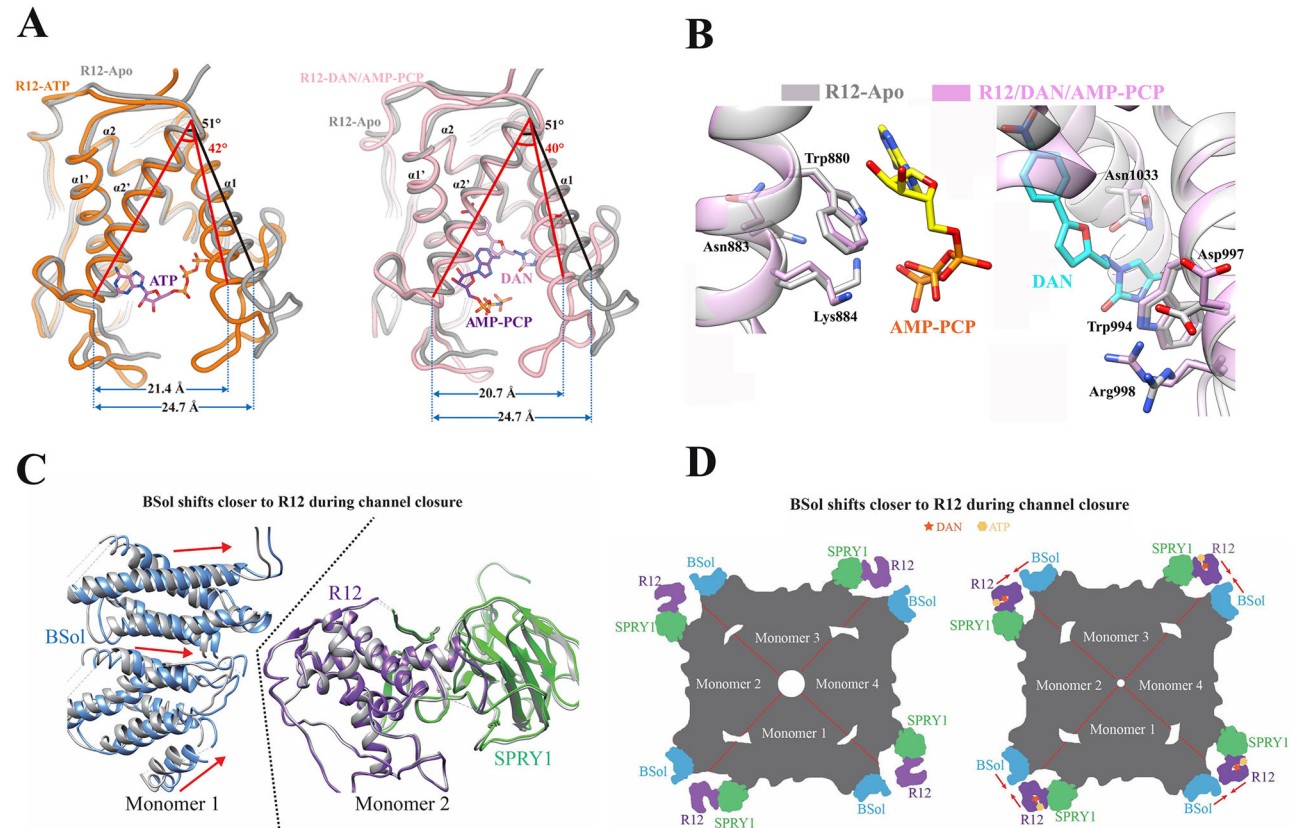

**Fig. 4 | Conformational changes in the R12 domain induced by ligand binding.**
**A** ATP binding induces closure of the domain, which is further closed upon DAN binding. The interhelical angle between helices α1 and α2', as well as the distance between the Cα atoms of Thr914 and Ile999, are used to quantify the conformational changes. **B** Superposition of RyR3 R12 Apo and RyR3 R12/DAN/AMP-PCP structures reveals side chain changes induced by AMP-PCP (left) and DAN (right). **C** Superposition of R12, SPRY1, and BSol domains isolated from the open (PDB ID 5TAL) and closed (PDB ID 5TAQ) cryo-EM structures of RyR1. R12 and SPRY1 move as rigid bodies, whereas BSol shifts closer to the R12-SPRY1 complex during channel closure. Ligand-induced conformational changes in R12 are expected to similarly reduce the distance between BSol and R12-SPRY1. **D** A schematic illustration showing that DAN and ATP binding induce conformational changes that stabilize the corner region of RyR1, promoting and stabilizing the closed state of the channel.

to high background binding of nucleotides. These findings underscore the distinct roles of Trp995 and Trp881 in ligand binding and highlight their critical contributions to the functional dynamics of the RyR1 R12 domain.

Interestingly, all DAN-contacting residues are conserved across the three isoforms of RyR (Fig. 2I). To explore how these ligands differentially modulate the isoforms, we focused on residues near the DAN binding site that are conserved between RyR1 and RyR3 but not between RyR1 and RyR2 (Fig. 2I). Substitutions were made to replace these residues with their RyR2 equivalents, and the effects on DAN binding were evaluated. Among the mutations, L969M and L983M completely abolish DAN binding, while others such as V974I, E1028D, and R1035K exhibited Kd values comparable to the WT (Fig. 3C and Supplementary Fig. 8C). Interestingly, both Leu969 and Leu983 are not directly located at the ligand binding site but instead in a peripheral region (Fig. 3D). These findings highlight the pivotal allosteric roles of these two residues in mediating the isoform-selective modulation of DAN.

**DAN/AZU induces R12 domain closure and allosteric modulation of gating**

To explore how DAN and AZU binding influence the RyR3 R12 domain structure, we compared the apo RyR3 R12 structure with both the RyR3 R12/ATP binary complex structure (PDB ID 6UHH) and our RyR3 R12/DAN/AMP-PCP ternary complex structure. Given that, the cryo-EM model and crystal structure of the apo R12 are highly

similar (Fig. 2B), we used the crystal structure for comparison due to its higher resolution. ATP binding alone induces significant clamshell-like closure of the R12 domain, decreasing the angle between the α1 and α2' helices by 9 degrees and reducing the distance between Thr914 and Ile999 by 3.3 Å. The binding of DAN or AZU further enhances this conformational shift, decreasing the angle by an additional 2 degrees and further reducing the distance between the two residues, leading to an even more pronounced closure of the clamshell (Fig. 4A).

Next, we leveraged our high-resolution structures to examine how ligand binding induces conformational changes in the coordinating residues. While both key tryptophan residues exhibit minimal changes upon ligand binding, several coordinating residues, including Lys884 (Arg885 in RyR1) and Asn1033 (Asn1034 in RyR1) undergo significant conformational rearrangements (Fig. 4B). These observations suggest that DAN and AMP-PCP binding follow an induced-fit mechanism, allowing the R12 domain to adapt dynamically to ligand interactions.

Finally, we examined how the conformational change in the R12 domain induced by DAN/AZU binding might influence channel gating. The R12 domain interacts with the SPRY1 domain from the same subunit and the BSol domain from the neighboring subunit (Fig. 4C). A comparison of the open-state (PDB ID 5TAL) and closed-state (PDB ID 5TAQ) cryo-EM structures of RyR1 revealed that the R12-SPRY1 domains move as a rigid body during gating, while the distance between the BSol domain and R12 is reduced, resulting in a more

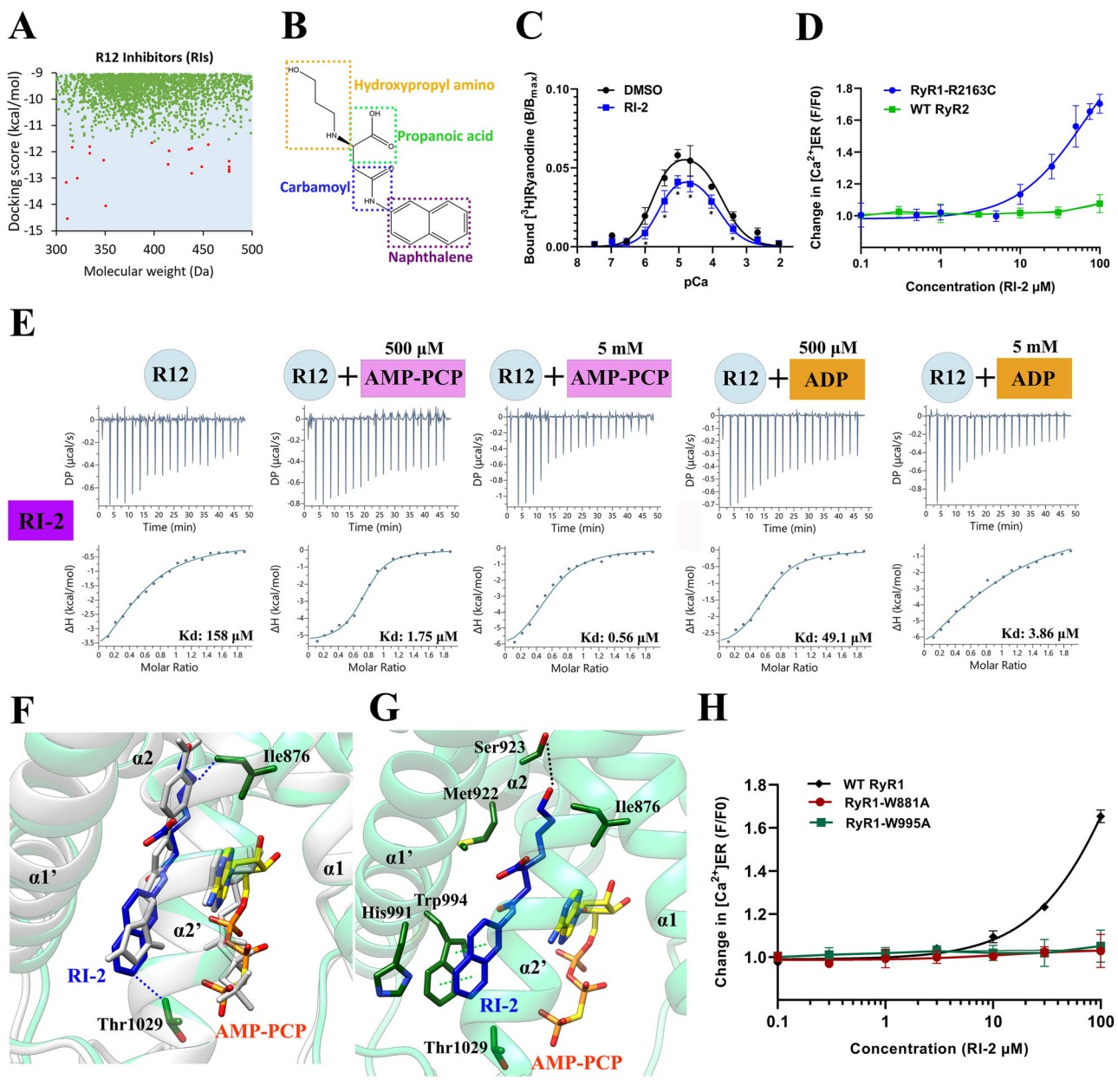

**Fig. 5 | Identification of potent RyR1 inhibitors targeting R12. A** Docking scores of the candidate compounds from a library of 1,270,000 molecules. The top 20 compounds, ranked by docking score, are highlighted in red, while the remaining compounds are shown in green. **B** Chemical structure of RI-2. **C** $Ca^{2+}$-dependent $[^3H]$ ryanodine binding to WT RyR1 in the presence of 100 uM RI-2. Data are presented as mean ± SD ($n = 4$ per group) and analyzed using a Student's t-test. Twotailed tests were used for all analyses. *$p < 0.05$ vs. DMSO. Source data are provided in the Source Data file. **D** Dose-dependent effect of RI-2 on ER luminal $[Ca^{2+}]$ changes in cell lines expressing RyR1 R2163C and WT RyR2. Data are presented as mean ± SD ($n = 3$ independent experiment). Source data are provided as a Source Data file. **E** ITC binding isotherms illustrating the interaction of RI-2 with the R12 domain of RyR1 in the absence or presence of AMP-PCP or ADP. The affinity and thermodynamic parameters are listed in Supplementary Table 1. **F** Superposition of the crystal structures of RyR3 R12/RI-2/AMP-PCP (light green) and RyR3 R12/DAN/AMP-PCP (white). Exclusive contacts with RI-2 are highlighted in green. **G** Enlarged view illustrating the ligand interactions with R12 in the RI-2 binding region. **H** Dose-dependent effect of RI-2 on ER luminal $[Ca^{2+}]$ changes in cell lines expressing WT RyR1, RyR1 W881A, and RyR1 W995A. Data are shown as the mean values ± SD ($n = 3$ independent experiment). To enhance the constitutive $Ca^{2+}$ leak via RyR1, the assay media was supplemented with 4 mM caffeine. RI-2 (0–100 μM) was added after 60 s. Source data are provided as a Source Data file.

compact conformation in this corner region. On the basis of these observations, we propose that DAN/AZU binding might induce a similar effect, making this region more compact, thereby stabilizing the closed state of the channel and inhibiting its activity (Fig. 4D).

### Identification of potent R12 domain-targeting Inhibitor
To develop a RyR1 inhibitor targeting its R12 domain, we performed virtual screening using Schrödinger, utilizing our high-resolution complex structure with pre-bound AMP-PCP as a template. From a library of 1,270,000 candidate compounds, we selected 20 top-ranking compounds based on their docking scores for further validation (Fig. 5A). Among these, 8 compounds demonstrated positive binding to the RyR1 R12 domain in the presence of AMP-PCP, exhibiting a broad range of binding affinities (Supplementary Fig. 9). To assess their functional impact, we conducted time-lapse $[Ca^{2+}]_{ER}$ measurements on MH mutant RyR1 R2163C. At 100 μM, three

compounds exhibited significant inhibitory effects compared the DMSO control (Supplementary Fig. 10). The most potent inhibitor, RI-2, contains a (hydroxypropyl)amino, a propanoic acid, a carbamoyl, and a naphthalene moiety (Fig. 5B). We further evaluated RI-2's impact on RyR1 channel function using [³H]ryanodine binding assays, demonstrating its inhibitory effect on WT RyR1 (Fig. 5C). Its IC$_{50}$ value was determined to be 28 μM through dose-dependent [Ca$^{2+}$]$_{ER}$ measurements, while no significant effect was observed on WT RyR2 (Fig. 5D). Similar to DAN and AZU, RI-2 binding to RyR1 R12 is AMP-PCP/ADP concentration-dependent, suggesting a similar binding site (Fig. 5E). Additionally, RI-2 exhibits isoform specificity comparable to DAN/AZU, binding RyR1 and RyR3 with similar affinities while showing no detectable binding to RyR2 (Supplementary Fig. 11).

To elucidate the precise mechanism of action of RI-2, we determined the crystal structure of RyR3 R12 in complex with RI-2 and AMP-PCP at a resolution of 2.50 Å (Supplementary Table 2). The structure revealed that AMP-PCP binds to the same site and adopts a similar pose as observed in the RyR3 R12/DAN/AMP-PCP complex, whereas RI-2 occupies the DAN-binding site but adopts a distinct conformation (Fig. 5F and Supplementary Fig. 7D). RI-2 binding is stabilized by π-π stacking between the naphthalene moiety of RI-2 and Trp994 (Trp995 in RyR1) on one side, and van der Waals interactions with AMP-PCP on the other (Fig. 5G). The binding mode of RI-2 was confirmed by ITC experiments, which showed that both W881A and W995A mutations affect its binding in a manner similar to that observed for DAN (Supplementary Fig. 12). In time-lapse ER Ca$^{2+}$ measurements, the inhibitory effect of RI-2 was completely lost in both mutations, confirming the importance of these two residues (Fig. 5H). Additionally, Ile876 (Ile877 in RyR1) and Thr1029 (Thr1030 in RyR1), which do not participate in DAN binding, interact with the the (hydroxypropyl)amino and naphthalene groups of RI-2 via van der Waals contacts, respectively. In addition, π-π stacking interaction observed between AMP-PCP and DAN is absent in the RI-2 complex structure (Fig. 5F). These distinct interactions suggest that although RI-2 engages the same binding site as DAN/AZU, it does so in a unique manner. Further optimization is needed to enhance its efficacy for potential therapeutic use in MH.

## Discussion

As the sole FDA-approved RyR-targeting drug, DAN has been used for decades for the treatment of MH, but its mechanism of action has long remained debated. Initial studies showed no effect of DAN on RyR1 reconstituted in planar lipid bilayers[38], whereas a later study showed it to only have an effect in the presence of CaM[14,38]. However, another study negated the requirement for CaM but instead pointed out that ATP was needed[13]. Our study shows that DAN is able to bind the R12 domain, albeit with weak affinity. However, in the presence of ADP or an ATP analog, its affinity increases significantly, and thus favors the finding by several studies[12,13,39,40]. We solved high-resolution crystal structures of DAN and its analog AZU, both in the presence of an adenine nucleotide, in complex with the RyR3 R12 domain (Fig. 2D-G, Supplementary Fig. 7). Both DAN and AZU share the same binding site deep in the cleft of the domain, with the difference in the head moiety facing the inside of the pocket. In addition, the binding of DAN/AZU is regulated by the presence of AMP-PCP or ADP, which occupies a complementary space in the same cavity, explaining why the DAN-binding affinity is AMP-PCP/ADP dependent.

A previous study on the cryo-EM structure of RyR1 in complex with ATP and ARM210, an RyR inhibitor currently in clinical trials, revealed that ATP binds to the R12 domain at a site similar to the DAN/AZU binding site observed in our structure[25,41,42]. In contrast, ARM210 binds to the AMP-PCP/ADP binding site identified in our structure[25]. This discrepancy may stem from differences in the properties of the two drugs, suggesting that ATP binding specificity to the R12 domain is relatively weak. This is supported by our ITC data, which show that upon introduction of the W881A mutation, nucleotides can bind to an

alternative site, competing with DAN binding. However, potential ambiguities in ATP modeling cannot be entirely ruled out, as the high flexibility of the corner region likely contributes to the relatively low local resolution in this part of the RyR1 cryo-EM structure. Additionally, another recent cryo-EM study modeled DAN in positions opposite to those observed in our crystal structure, likely due to low local resolution in that study (Fig. 2H)[26]. During writing of this manuscript, a preprint was published reporting two cryo-EM structures of RyR1 bound to DAN with ATP or ADP[43]. Their findings support our model of DAN and nucleotide binding sites, along with the clamshell-like conformational changes induced upon binding. However, this study did not address the molecular basis of DAN's isoform selectivity in the context of the full-length receptor. Due to the peripheral location and inherent flexibility of R12, structural characterization of this domain in the full-length RyR1 remains challenging, limiting resolution and hindering confident assignment of small-molecule binding. While the crystallographic studies of the isolated R12 domain provide atomic-level detail, the full protein context is absent, potentially overlooking allosteric and membrane-associated effects. Therefore, integrative approaches are needed to bridge domain-level and full-length structural insights and to fully elucidate both binding interactions and functional mechanisms.

Whereas both RyR1 and RyR3 can be effectively inhibited by DAN, it is still controversial whether RyR2 is responsive to DAN. Our ITC results indicate the RyR isoform selectivity of DAN/AZU. Combined with the mutagenesis results, we identified Leu969 and Leu983 as the key residues for this selectivity (Fig. 3C-D). Unexpectedly, these two residues are not located directly at the DAN/AZU binding site but instead at a buried position; therefore, the substitutions might affect the conformation of the domain and allosterically affect DAN/AZU binding. The R12 domain is involved in contact with the neighboring tetramer RyR. The difference in the quaternary configurations of different isoforms might also contribute to the binding of ligands targeting this domain.

Our structure reveals the binding site of DAN, providing a foundation for structure-based screening and the development of potent compounds to treat MH and other RyR-associated diseases. Under physiological condition, ATP and ADP are present in the millimolar range in the cytosol of muscle cells. Given the significantly higher affinity of DAN in the presence of nucleotides, it is likely that DAN primarily binds as a ternary complex with nucleotides, making it challenging to use the apo-state cryo-EM or crystal structures for accurate drug design. Our high-resolution crystal structures of R12 in complex with DAN/AZU and AMP-PCP/ADP provide a precise structural template, enabling the identification of inhibitors with enhanced solubility that efficiently and selectively inhibit RyR1. Additionally, the ternary complex structure of R12 with RI-2 and AMP-PCP offers valuable insights for further compound optimization. This iterative approach will be applied to develop more potent therapeutic molecules. Future designs could also explore linking the nucleotide structures to DAN, AZU, or RI-2 to further enhance its binding affinity and specificity.

## Methods

### Protein expression and purification

The R12 domain from three human RyR isoforms (RyR1 854–1054, RyR2 865–1065, RyR3 853–1053) was cloned into a modified pET28 vector containing an N-terminal 6xHis tag, an MBP-fusion protein, and a TEV cleavage site. The plasmids were subsequently transformed into *E. coli* BL21 (DE3) cells (NEB, USA) and cultured in *Luria-Bertani* (LB) media at 37 °C. Protein expression was induced with 0.4 mM isopropyl β-D-thiogalactoside (IPTG) when the optical density (OD)$_{600}$ reached ~0.8, followed by incubation at 18 °C for 12 h. The cells were harvested via centrifugation at 5,000 × g for 15 min at 4 °C and homogenized in Buffer A (10 mM HEPES, pH 7.4, 250 mM KCl) supplemented with

25 µg/mL DNaseI, 25 µg/mL lysozyme, 1 mM phenylmethylsulfonyl fluoride (PMSF), and 20 mM imidazole. The homogenate was centrifuged at 12,000 × g for 45 min, and the supernatant was filtered through a syringe filter (0.22 µm).

Proteins were initially purified using a Ni-NTA gravity column (GenScript), washed with 10 column volumes of Buffer A, and eluted with Buffer A containing 500 mM imidazole. The eluate was dialyzed against 2 L of Buffer A for 4 h to remove imidazole and then loaded onto an amylose resin column (New England Biolabs). The collected protein was cleaved with recombinant TEV protease at a 1:30 molar ratio during overnight dialysis against 2 L of Buffer A. The dialyzed sample was then passed through a HisTrap HP column (GE Healthcare) to remove the His-MBP-tag. The flow-through containing the cleaved R12 protein was collected and dialyzed against 2 L of Buffer B (50 mM Tris–HCl, pH 6.8, 1 mM β-mercaptoethanol) for 3 h. Further purification was performed using an SP Sepharose high-performance column (GE Healthcare) with a linear gradient from 20 500 mM KCl in elution buffer (10 mM Tris–HCl, pH 6.8, and 1 mM β-mercaptoethanol). The purified protein was concentrated via Amicon concentrators (10 K MWCO, Millipore) and subjected to size-exclusion chromatography (SEC) on a Superdex 200 16/600 gel-filtration column (GE Healthcare), with Buffer A used as the elution buffer. The protein purity was examined on a 15% (w/v) SDS polyacrylamide gel. The final protein sample was concentrated to 20 mg/ml, exchanged into crystallization buffer (10 mM HEPES pH 7.4, 50 mM KCl, and 1 mM tris(2-carboxyethyl)phosphine)), and stored at −80 °C.

### Isothermal titration calorimetry

Dantrolene (F368) and Azumolene (EU4093) were purchased from MedChemExpress (MCE). Purified R12 domain protein from the three RyR isoforms, fused with hexahistidine-tagged MBP (HMT-R12) was dialyzed overnight at 4 °C in ITC buffer (200 mM HEPES, pH 7.4, 150 mM KCl). ITC experiments were performed using a PEAQ-ITC instrument (Malvern, UK) at 25 °C with a stirring speed of 750 rpm. Each titration consisted of 19 consecutive injections of 2 µL titrant containing 500 µM DAN/AZU or AMP-PCP/ADP into the cell containing 50 µM HMT-R12. For the ternary binding experiments, 500 µM or 5 mM AMP-PCP/ADP was preincubated with both the titrants and titrates. Control experiments involved titrating 500 µM DAN/AZU or AMP-PCP/ADP into 50 µM hexahistidine-tagged MBP without R12, which yielded heat signals indistinguishable from those of the buffer injections (Supplementary Fig. 13). The data was fit to a one-site fitting model after background buffer subtraction. The RyR1 R12 mutants were generated using a QuikChange mutagenesis kit (Stratagene) according to the manufacturer's protocol, with primer sequences listed in Supplementary Table 3.

### Time-lapse [Ca$^{2+}$]$_{ER}$ measurement

HEK293 cells stably expressing R-CEPIA1er and RyRs (WT RyR1, RyR1-W881A, RyR1-W995A, RyR1-R2163C, and WT RyR2)[26,27] were used in this study. Briefly, the cells were seeded at a density of $2 \times 10^4$ cells/well in a 96-well black, clear-bottom plate (Corning). Doxycycline (2 µg/mL) was added 12 h after seeding, and the medium was replaced 24 h later with HEPES-buffered Krebs solution (5 mM HEPES, pH 7.4; 140 mM NaCl; 5 mM KCl; 2 mM CaCl$_2$; 1 mM MgCl$_2$; and 11 mM glucose). Fluorescence measurements were conducted using a FlexStation 3 fluorometer (Molecular Devices) at 37 °C. The R-CEPIA1er was excited at 560 nm, emitted at 610 nm, and captured every 10 seconds over 300 seconds. Different concentrations of compound, dissolved in buffer containing 1% DMSO, were added 100 s after recording began. The fluorescence changes ($F/F_0$) were calculated by dividing the average fluorescence of the last 100 seconds ($F$) by that of the first 100 seconds ($F_0$). All measurements were performed in triplicate.

### [³H]Ryanodine binding

The assay was performed as previously described[44,45] with minor modifications. Briefly, microsomes from HEK293 cells stably expressing WT RyR1 were incubated with 5 nM [³H]ryanodine for 5 h at 25 °C in a medium containing 0.17 M NaCl, 20 mM MOPSO (pH 7.0), 2 mM dithiothreitol, 1 mM β,γ-methyleneadenosine 5′-triphosphate, and free Ca$^{2+}$ buffered with 10 mM EGTA at varying concentrations. Free Ca$^{2+}$ levels were calculated using WEBMAXC STANDARD[46]. Protein-bound [³H]Ryanodine ($B$) was isolated by filtration through polyethyleneimine-treated GF/B filters (Whatman) using a Micro 96 Cell Harvester (Skatron Instruments). Nonspecific binding was assessed by adding 20 µM unlabeled ryanodine. The maximum binding capacity of [³H]ryanodine ($B_{max}$) was determined via Scatchard plot analysis with various concentrations (3–20 nM) of [³H]ryanodine in a high-salt medium containing 1 M NaCl. The resulting $B/B_{max}$ values represent the average activity of the cells.

### Crystallization, data collection, and structure determination

The apo and complex protein crystals were screened at a protein concentration of approximately 20 mg/mL using the hanging-drop method at 18 °C. For the ternary complexes, the molar ratio of hRyR3-R12 to DAN/AZU and AMP-PCP/ADP was 1:2:10. The mixtures were incubated at 4 °C for 1 hour prior to initial crystallization screening, which was performed via the sitting–drop vapor–diffusion method with commercial sparse matrix screening kits from Hampton Research and Molecular Dimensions. Crystallization was performed in a 96-well format at a 1:1 protein-to-reservoir ratio, facilitated by an automated liquid handling system (Gryphon, Art Robbins). Following the identification of initial hits, the crystallization conditions were optimized via the hanging-drop vapor–diffusion method in a 24-well format. The optimal crystallization conditions for RyR3 R12 apo include 0.1 M Bis-Tris pH 6.5, and 28% PEG2000; for RyR3 R12/DAN/AMP-PCP, 0.7 M sodium acetate trihydrate and 14% PEG3350; for RyR3 R12/DAN/ADP, 0.6 M sodium acetate trihydrate and 18% PEG3350; for RyR3 R12/AZU/AMP-PCP, 0.4 M sodium chloride, 0.1 M HEPES pH 7.5, and 20% PEG3350, and for RyR3 R12/RI-2/AMP-PCP, 0.1 M Tris:HCl pH 8.5, 20% PEG 1000, and 10% glycerol. For RyR3 R12 apo and RyR3 R12/RI-2/AMP-PCP, 25% Glycerol was used as a cryoprotectant, while 25% PEG200 was used for RyR3 R12/DAN/AMP-PCP, RyR3 R12/DAN/ADP, and RyR3 R12/AZU/AMP-PCP.

Diffraction data were collected on BL18U1 and BL10U2 at the Shanghai Synchrotron Radiation Facility (SSRF) at resolutions of 1.97 Å for RyR3 R12 apo, 2.79 Å for RyR3 R12/DAN/AMP-PCP, 2.84 Å for RyR3 R12/DAN/ADP, 3.00 Å for RyR3 R12/AZU/AMP-PCP, and 2.50 Å for RyR3 R12/RI-2/AMP-PCP. Data processing, including indexing, integration, and scaling, was performed using HKL2000. Molecular replacement was performed using PHENIX (version 1.18.2–3874)[47]. The structure of 6UHA was used as the template for RyR3 R12 apo, while 6UHH was used for RyR3 R12/DAN/AMP-PCP. The remaining structures, including RyR3 R12/DAN/ADP, RyR3 R12/AZU/AMP-PCP, and RyR3 R12/RI-2/AMP-PCP, were modeled using 9L90 as the template. The structural model was subsequently built using COOT (version 0.9.8.93)[48]. Refinement was performed in PHENIX (version 1.18.2–3874)[47], and structural figures were generated via UCSF Chimera (version 1.14)[49]. Interaction analyses were conducted using the Discovery Studio interaction analysis module (Accelrys, USA). The statistics related to data collection and the final refinement process are summarized in Supplementary Table 2.

### Homology model

The homology models of the R12 domain of RyR1 and RyR2 with DAN and AMP-PCP (Supplementary Data 1 and 2, respectively) were built using our crystal structure of RyR3 R12/DAN/AMP-PCP (PDB ID 9L90) as a template with the Prime module (Prime, Schrodinger, LLC, New

York, NY, 2024). The generated models were refined and energy minimized via the OPLS 2005 force field[50].

## Virtual screening

A total of 1,270,000 ligands from the Chinese National Compound Library (CNCL) (*24/12/05*) were prepared using the LigPrep module in the Schrödinger suite 2017[50]. The two-dimensional (2D) structures of the ligands were converted into energy-minimized three-dimensional (3D) structures using the OPLS3 force field, and multiple conformers were generated through a rapid torsion search performed by MacroModel[50,51]. During ligand preparation, the pH was set to $7.0 \pm 2.0$ and and the Epik module was used to generate tautomeric and ionization states[50]. The protein structure was prepared using the protein preparation wizard in Maestro (version 11.1)[50], and energy-minimized with the OPLS3 force field. A grid file was generated using the Receptor Grid Generation module in Glide (version 7.4), with the grid box centered on Trp995 in the presence of AMP-PCP at Trp881. Ligands were docked within a 20 Å radius of this centroid.

Virtual screening was conducted to identify hit compounds capable of inhibiting RyR1. Prior to screening, the drug-likeness of the compounds was assessed using Lipinski's rule of five to filter the library. The Glide docking program (version 7.4) was employed for virtual screening, which was carried out in a hierarchical manner: High-Throughput Virtual Screening (HTVS), followed by Standard Precision (SP), and finally Extra Precision (XP) docking[50]. Molecules exhibiting the highest Glide scores and Glide energies were visually inspected and prioritized for further analysis.

## Reporting summary

Further information on research design is available in the Nature Portfolio Reporting Summary linked to this article.

## Data availability

The diffraction data and atomic coordinates have been deposited in the PDB under the following accession codes 9L92 (RyR3 R12 apo), 9L90 (RyR3 R12/DAN/AMP-PCP), 9L9B (RyR3 R12/DAN/ADP), 9L91 (RyR3 R12/AZU/AMP-PCP), and 9LS7 (RyR3 R12/RI-2/AMP-PCP). The structures used in this paper are available in the PDB database under accession codes 6UHA, 6UHH, 6M2W, 9C1E, 9CGP, 5TAL, and 5TAQ.

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

## Acknowledgements

This research was funded by the National Key Research and Development Program of China (nos. 2022YFE0108400 and 2025YFC3409400 to Z.Y.), the National Natural Science Foundation of China (no. 32372580 to Z.Y.), the Haihe Laboratory of Sustainable Chemical Transformations (no. 24HHWCSS00005 to Z.Y.), and the Emerging Frontiers Cultivation Program of Tianjin University Interdisciplinary Center (to Z.Y.). We thank the staff from the beamline BL18U1 and BL10U2 at the Shanghai Synchrotron Radiation Facility (SSRF).

## Author contributions

Conceptualization: H.H. and Z.Y.; Methodology: H.H., Y.Y., T.M., F.V.P., and Z.Y.; Investigations: H.H., L.L., Z.W., R.S., Q.W., X.L., N.K., T.K., T.Y., Y.S.C., W.W., H.Z., T.M., and Z.Y.; Resources: Z.Y.; Data analysis: H.H., L.L., and T.M.; Writing – original draft: H.H. and Z.Y.; Writing – review and editing: H.H., Y.Y., T.M., F.V.P., and Z.Y.; Supervision: Z.Y.; Funding acquisition: Z.Y.

## Competing interests

The authors declare no competing interests.
