## [Transparent Peer Review file · Nature Communications]

Crystal structures of Ryanodine Receptor reveal dantrolene and azumolene interactions guiding inhibitor development

Corresponding Author: Professor Zhiguang Yuchi

Version 0:

Reviewer comments:

Reviewer #1

(Remarks to the Author)

The ms by Hadiatullah et al examines the conformation of Repeat 12 (R12) domain of the ryanodine receptor (RyR) using X-ray crystallography. The authors studied the binding affinity of the R12 domain with two inhibitors, dantrolene (DAN) and azumolene (AZU). DAN is the only FDA-approved drug for malignant hypertrophy (MH), but its hepatotoxicity prevents its usage as a long-term drug.

The authors co-crystallized the R12 domain with DAN or AZU in the presence of AMP-PCP or ADP. They also studied the binding affinity of the inhibitors to the R12 domain in the presence or absence of AMP-PCP/ADP. Among the isoforms, RyR1 and RyR3 have sequence similarity and, unlike RyR2, show strong binding affinity to DAN and AZU. The authors could obtain co-crystals only for the RyR3 R12 domain. The authors also generated functional mutants to test residues essential for the binding of DAN and AZU. Lastly, they used Schrodinger software to identify novel inhibitors against the R12 domain, with RI-2 identified as the most potent inhibitor among those tested. Co-crystallization of RI-2 with RyR3 R12 domain and AMP-PCP was performed to study the binding site. RI-2 was found to have a similar binding site as DAN/AZU but with slightly different coordination compared to the other inhibitors.

The ms is generally well written. The structural basis of inhibitor binding in RyR channels is likely to be of interest to a broad audience in structural biology, pharmacology, cell biology, and channel biophysics. I have a few comments that the authors may wish to address to improve the presentation.

1. There seems to be some information missing from Table 2. Specifically, from the data section, there are no hi-res shell values provided for Rmerge, Rmeas, Rpim, CC1/2, Completeness, or Mean I/sigma. Without this information it is difficult for readers to evaluate the basis for the resolution cut-off imposed by the authors, so it should be included. Additionally, the Rfree and Rwork values for refinement appear to be very high, greater than 29% for Rwork and greater than 35% for Rfree for all but the R12 apo structure. This suggests that the refinement parameters have not been optimized, or substantial errors may remain in the model. The authors should perform further refinement and/or rebuilding to remedy these errors.

2. The authors may wish to calculate omit maps to confirm the locations of the drug binding sites and poses.

3. Lines 132 and 353. Strictly speaking, it is unlikely that Kd values determined from ITC experiments with C<5 are very meaningful on a quantitative basis, so the interpretation of these estimated Kd values should be moderated in the text.

Reviewer #2

(Remarks to the Author)

Dantrolene is the only approved inhibitor of RyR-mediated Ca²⁺ release from the sarcoplasmic reticulum (SR) for treating malignant hyperthermia (MH) and central core disease (CCD). While the 3D structures of full-length ryanodine receptors have been resolved, the exact binding site and mechanism of dantrolene on RyRs remain poorly understood. A recent cryo-EM study mapped the dantrolene binding site to the P1 domain of RyR1 (ref #26), but its resolution is limited, leaving uncertainties.

This study reports the crystal structure of the R12 domain, located in a similar region as the P1 domain, at significantly higher resolution, enabling precise localization of the dantrolene binding site. The authors convincingly demonstrate that dantrolene and its analog, azumolene, bind to a cavity within the R12 domain. Key findings include the enhancement of dantrolene binding affinity by adenine nucleotides (ADP or AMP-PCP) through binding at a site near the dantrolene binding site and the insensitivity of RyR2 to dantrolene due to leucine-to-methionine substitutions at two critical residue positions in RyR2. Additionally, this study clarifies that a tryptophan residue (W881), previously modeled near dantrolene (Iyer et al., 2024), actually binds adenine nucleotides.

The resolved 3D structure of the R12 domain is robust, supported by rigorous isothermal titration calorimetry (ITC) assays validating functional effects of ligand binding and amino acid mutations. Although the study used the isolated R12 domain of RyR3 (which is not directly implicated in MH/CCD) for the crystal structure analysis, this does not appear to be a significant limitation, as the dantrolene-binding residues identified in the R12 domain of RyR3 are highly conserved in RyR1, and both RyR1 and RyR3 are sensitive to dantrolene. Furthermore, this study identifies a RyR1 inhibitor through R12 structure-guided virtual screening, demonstrating the potential utility of the R12 domain structure for developing new RyR inhibitors for clinical applications.

Major Comments:

1. The tryptophan residues described in the main text (W881 and W995) appear to be mislabeled as W880 and W994 in the ribbon diagrams in Figure 2.
2. The main text states, "All DAN-contacting residues are conserved across the three isoforms of RyR (Figure 2G)" (lines 296-297), but Figure 2G only compares two RyR3 structures (one bound to dantrolene and the other to azumolene).
3. Unlike this study, two closely related studies (refs #26 & #44) determined the dantrolene binding site using full-length RyR1. Please discuss the advantages and potential limitations of using the isolated R12 domain of RyR3 in this study.
4. When citing the recent bioRxiv paper (ref #44), the authors state that "Their findings support our model regarding the binding sites of DAN/AZU and the nucleotides." It would be beneficial to describe the similarities and differences in major findings between these studies. For example, did the other study investigate why dantrolene does not bind to RyR2?
5. An amino acid sequence alignment of the R12 domain across RyR1, RyR2, and RyR3 would help readers understand the rationale for testing L969M and L983M mutations. Since L969 and L983 are not the only potential candidate residues for these experiments, the authors likely tested additional mutations that showed no effect. Please include these data in the supplementary materials.

Minor Issues:

1. Both this study and ref #26 mapped the dantrolene binding site to a similar location but referred to the binding domain differently: P12 in this study and P1 in ref #26. This could be confusing for readers. Please briefly explain the similarities and differences between these two domains and why they are named differently.
2. The yellow text labels in Figures 2, 3, 4, and 5 are difficult to read. Use a higher-contrast color.
3. Define "ITC" as "isothermal titration calorimetry" at first mention (currently abbreviated ambiguously as "Calorimetric (ITC)" in line 34).
4. Briefly describe the rationale for using AMP-PCP instead of ATP in the experiments.
5. All amino acid residues, including those in the R12 domain of RyR3, are numbered according to their positions in RyR1. Please clarify this in the main text following the description of the R12 domain of RyR3 used for the crystal structure study.

Reviewer #3

(Remarks to the Author)

In this study, the crystal structure of RyR complexed with ligands was analyzed to elucidate its inhibitory mechanism. Two tryptophan residues that form key stacking interactions with ligands were identified. A hierarchical virtual screening strategy (HTVS → SP → XP) was applied to screen a library of 1.27 million compounds to identify potential candidates. Docking results were validated using ITC and calcium imaging assays. Further investigations into receptor flexibility may be needed to fully characterize the inhibitory mechanism. Techniques such as Induced-Fit Docking (IFD) or Molecular Dynamics (MD) simulations could be employed to explore conformational changes within the binding pocket.

Version 1:

Reviewer comments:

Reviewer #1

(Remarks to the Author)

The revised ms by Hadiatullah et al examines the conformation of Repeat 12 (R12) domain of the ryanodine receptor (RyR), and binding affinity of the R12 domain with two inhibitors, dantrolene (DAN) and azumolene (AZU). The ms is generally well written, and the structural basis of inhibitor binding in RyR channels is likely to be of interest to a broad audience in structural biology, pharmacology, cell biology, and channel biophysics.

I am satisfied with the authors revisions and I have no further criticisms.

Reviewer #2

(Remarks to the Author)

The authors have addressed all of my comments appropriately.

I only have a minor suggestion about a newly added paragraph for the authors to consider:

Changing a few word in this sentence "While the crystallographic studies of isolated R12 domain provide atomic-level detail, they lack the full protein context, potentially overlooking allosteric and membrane-associated effects." so that it becomes "While the crystallographic studies of the isolated R12 domain provide atomic-level detail, the full protein context is absent, potentially overlooking allosteric and membrane-associated effects." I think such a change can improve logical flow.

Reviewer #3

(Remarks to the Author)

The revised manuscript has addressed my concerns. Therefore, it is recommended that the manuscript be accepted for publication.

We thank all the reviewers for the constructive comments and suggestions, which have led to an overall improvement in our efforts to communicate the results. I have reproduced your comments verbatim below together with our responses. For clarity, your criticisms and/or suggestions are coloured in dark red, while our response is shown in dark blue.

Reviewer #1 (Remarks to the Author):

The ms by Hadiatullah et al examines the conformation of Repeat 12 (R12) domain of the ryanodine receptor (RyR) using X-ray crystallography. The authors studied the binding affinity of the R12 domain with two inhibitors, dantrolene (DAN) and azumolene (AZU). DAN is the only FDA-approved drug for malignant hypertrophy (MH), but its hepatotoxicity prevents its usage as a long-term drug.

The authors co-crystallized the R12 domain with DAN or AZU in the presence of AMP-PCP or ADP. They also studied the binding affinity of the inhibitors to the R12 domain in the presence or absence of AMP-PCP/ADP. Among the isoforms, RyR1 and RyR3 have sequence similarity and, unlike RyR2, show strong binding affinity to DAN and AZU. The authors could obtain co-crystals only for the RyR3 R12 domain. The authors also generated functional mutants to test residues essential for the binding of DAN and AZU. Lastly, they used Schrodinger software to identify novel inhibitors against the R12 domain, with RI-2 identified as the most potent inhibitor among those tested. Co-crystallization of RI-2 with RyR3 R12 domain and AMP-PCP was performed to study the binding site. RI-2 was found to have a similar binding site as DAN/AZU but with slightly different coordination compared to the other inhibitors.

The ms is generally well written. The structural basis of inhibitor binding in RyR channels is likely to be of interest to a broad audience in structural biology, pharmacology, cell biology, and channel biophysics. I have a few comments that the authors may wish to address to improve the presentation.

COMMENT #1: There seems to be some information missing from Table 2. Specifically, from the data section, there are no hi-res shell values provided for Rmerge, Rmeas, Rpim, CC1/2, Completeness, or Mean I/sigma. Without this information it is difficult for readers to evaluate the basis for the resolution cut-off imposed by the authors, so it should be included. Additionally, the Rfree and Rwork values for refinement appear to be very high, greater than 29% for Rwork and greater than 35% for Rfree for all but the R12 apo structure. This suggests that the refinement parameters have not been optimized, or substantial errors may remain in the model. The authors should perform further refinement and/or rebuilding to remedy these errors.

RESPONSE #1: We thank the reviewer for this valuable comment. We have updated Supplementary Table 2 to include all high-resolution shell statistics, including Rmerge, CC1/2, completeness, and Mean I/ σ . In addition, we have conducted multiple rounds of model rebuilding and refinement, which have substantially improved the Rwork and Rfree values across all structures. The updated refinement statistics are included in the revised Supplementary Table 2.

COMMENT #2: The authors may wish to calculate omit maps to confirm the locations of the drug binding sites and poses.

RESPONSE #2: We thank the reviewer for this helpful suggestion. To validate the identified drug-binding sites and ligand conformations, we performed composite omit map calculations using Phenix. Representative omit maps have been provided for each structure in the new Supplementary Figure S7.

COMMENT #3: Lines 132 and 353. Strictly speaking, it is unlikely that K_d values determined from ITC experiments with $C < 5$ are very meaningful on a quantitative basis, so the interpretation of these estimated K_d values should be moderated in the text.

RESPONSE #3: We appreciate the reviewer's insightful comment. In response, we have moderated our interpretation and removed the K_d values derived from ITC measurements as follows:

1. Line 109–110: “Both DAN and AZU exhibited clear binding to the R12 domain of RyR1 with comparable apparent binding affinities.”
2. Line 322–324: “Among these, 8 compounds demonstrated positive binding to the RyR1 R12 domain in the presence of AMP-PCP, exhibiting a broad range of binding affinities.”

Reviewer #2 (Remarks to the Author):

Dantrolene is the only approved inhibitor of RyR-mediated Ca^{2+} release from the sarcoplasmic reticulum (SR) for treating malignant hyperthermia (MH) and central core disease (CCD). While the 3D structures of full-length ryanodine receptors have been resolved, the exact binding site and mechanism of dantrolene on RyRs remain poorly understood. A recent cryo-EM study mapped the dantrolene binding site to the P1 domain of RyR1 (ref #26), but its resolution is limited, leaving uncertainties.

This study reports the crystal structure of the R12 domain, located in a similar region as the P1 domain, at significantly higher resolution, enabling precise localization of the dantrolene binding site. The authors convincingly demonstrate that dantrolene and its analog, azumolene, bind to a cavity within the R12 domain. Key findings include the enhancement of dantrolene binding affinity by adenine nucleotides (ADP or AMP-PCP) through binding at a site near the dantrolene binding site and the insensitivity of RyR2 to dantrolene due to leucine-to-methionine substitutions at two critical residue positions in RyR2. Additionally, this study clarifies that a tryptophan residue (W881), previously modeled near dantrolene (Iyer et al., 2024), actually binds adenine nucleotides. The resolved 3D structure of the R12 domain is robust, supported by rigorous isothermal titration calorimetry (ITC) assays validating functional effects of ligand binding and amino acid mutations. Although the study used the isolated R12 domain of RyR3 (which is not directly implicated in MH/CCD) for the crystal structure analysis, this does not appear to be a significant limitation, as the dantrolene-binding residues identified in the R12 domain of RyR3 are highly conserved in RyR1, and both RyR1 and RyR3 are sensitive to dantrolene. Furthermore, this study identifies a RyR1 inhibitor through R12 structure-guided virtual screening, demonstrating the potential utility of the R12 domain structure for developing new RyR inhibitors for clinical applications.

Major Comments:

COMMENT #1: The tryptophan residues described in the main text (W881 and W995) appear to be mislabeled as W880 and W994 in the ribbon diagrams in Figure 2.

RESPONSE #1: We apologize for the confusion. In Fig. 2, residue numbering follows the RyR3 R12 domain, while the main text refers to RyR1 R12 numbering. To prevent ambiguity, we have now included the corresponding RyR1 residue numbers in brackets throughout the main text for clarity. Additionally, in the legend of Fig. 2, we mentioned “All residue numbers are based on RyR3 sequence numbering.”

COMMENT #2: The main text states, “All DAN-contacting residues are conserved across the three isoforms of RyR (Figure 2G)” (lines 296-297), but Figure 2G only compares two RyR3 structures (one bound to dantrolene and the other to azumolene).

RESPONSE #2: Thank you for pointing this out. The correct figure reference is Fig. 2I, which shows the sequence alignment across the three RyR isoforms. We have corrected the citation in the revised text accordingly.

COMMENT #3: Unlike this study, two closely related studies (refs #26 & #44) determined the dantrolene binding site using full-length RyR1. Please discuss the advantages and potential limitations of using the isolated R12 domain of RyR3 in this study.

RESPONSE #3: We appreciate this thoughtful suggestion. In response, we have expanded the discussion to include the strengths and limitations of using the isolated R12 domain:

Lines 392–399: “Due to the peripheral location and inherent flexibility of R12, structural characterization of this domain in the full-length RyR1 remains challenging, limiting resolution and hindering confident assignment of small-molecule binding. By contrast, crystallographic studies of isolated domains provide atomic-level detail but lack the full protein context, potentially overlooking allosteric and membrane-associated effects. Therefore, integrative approaches are needed to bridge domain-level and full-length structural insights and to fully elucidate both binding interactions and functional mechanisms.”

COMMENT #4: When citing the recent bioRxiv paper (ref #44), the authors state that “Their findings support our model regarding the binding sites of DAN/AZU and the nucleotides.” It would be beneficial to describe the similarities and differences in major findings between these studies. For example, did the other study investigate why dantrolene does not bind to RyR2?

RESPONSE #4: Thank you for the helpful suggestion. We have revised the text to better articulate the overlap in findings: Lines 389–392: “Their findings support our model of DAN and nucleotide binding sites, along with the clamshell-like conformational changes induced upon binding. However, this study did not address the molecular basis of DAN's isoform selectivity in the context of the full-length receptor.”

COMMENT #5: An amino acid sequence alignment of the R12 domain across RyR1, RyR2, and RyR3 would help readers understand the rationale for testing L969M and L983M mutations. Since L969 and L983 are not the only potential candidate residues for these experiments, the authors

likely tested additional mutations that showed no effect. Please include these data in the supplementary materials.

RESPONSE #5: We appreciate the reviewer's thoughtful suggestion and have addressed it by including: (1) A sequence alignment of the R12 domain across RyR1, RyR2, and RyR3, highlighting the selection of isoform-selective residues (Fig. 2I); (2) A structural superposition of the R12 domains from RyR1 and RyR2, illustrating the locations of these isoform-selective residues relative to the ligand-binding site (Fig. 3D); (3) ITC results of three additional mutations (V974I, E1028D, and R1035K) demonstrating similar DAN-binding as the WT (Supplementary Figure S8).

Minor Issues:

COMMENT #1: Both this study and ref #26 mapped the dantrolene binding site to a similar location but referred to the binding domain differently: P12 in this study and P1 in ref #26. This could be confusing for readers. Please briefly explain the similarities and differences between these two domains and why they are named differently.

RESPONSE #1: Thank you for your comment. Both R12 and P1 refer to the same region within RyR. The difference in naming arises from historical conventions. This domain consists of two tandemly repeated sequences within the RyR protein, forming a structure with pseudo two-fold symmetry. Some studies refer to the entire tandem unit as a single domain (e.g., P1 in ref #26), while others like ours treat the repeats as a combined unit and refer to it as Repeat12 (R12). Despite the naming variation, both terms describe the same structural region. The same applies to P2 and R34, which also refer to the same domain.

COMMENT #2: The yellow text labels in Figures 2, 3, 4, and 5 are difficult to read. Use a higher-contrast color.

RESPONSE #2: We have revised Fig. 2, 3, 4, and 5 by replacing the yellow text with a higher-contrast color to improve readability.

COMMENT #3: Define “ITC” as “isothermal titration calorimetry” at first mention (currently abbreviated ambiguously as “Calorimetric (ITC)” in line 34).

RESPONSE #3: We have revised the abstract and main text to define ITC as “isothermal titration calorimetry” upon first mention, in line with the reviewer’s suggestion.

COMMENT #4: Briefly describe the rationale for using AMP-PCP instead of ATP in the experiments.

RESPONSE #4: Thank you for the reminder. We have clarified upon first mention in the Introduction that AMP-PCP is a non-hydrolyzable ATP analog. It is commonly used to prevent nucleotide hydrolysis during structural and binding studies.

COMMENT #5: All amino acid residues, including those in the R12 domain of RyR3, are numbered according to their positions in RyR1. Please clarify this in the main text following the description of the R12 domain of RyR3 used for the crystal structure study.

RESPONSE #5: Thank you for the helpful suggestion. We have clarified this point in the main text after introducing the RyR3 R12 domain used for crystallography, adding the following sentence:

Lines 158–160: “Throughout the text, residue numbers follow RyR1 numbering, except in the RyR3 structural descriptions, where the corresponding RyR1 residues are indicated in brackets.”

Reviewer #3 (Remarks to the Author):

COMMENT #1: In this study, the crystal structure of RyR complexed with ligands was analyzed to elucidate its inhibitory mechanism. Two tryptophan residues that form key stacking interactions with ligands were identified. A hierarchical virtual screening strategy (HTVS → SP → XP) was applied to screen a library of 1.27 million compounds to identify potential candidates. Docking results were validated using ITC and calcium imaging assays. Further investigations into receptor flexibility may be needed to fully characterize the inhibitory mechanism. Techniques such as Induced-Fit Docking (IFD) or Molecular Dynamics (MD) simulations could be employed to explore conformational changes within the binding pocket.

RESPONSE #1: Thank you for the constructive suggestion. To assess the potential impact of ligand binding on receptor dynamics, we performed 100-ns molecular dynamics (MD) simulations on four systems: R12 alone, R12/AMP-PCP, R12/DAN/AMP-PCP, and R12/RI-2/AMP-PCP. Although the apo R12 structure required a longer equilibration period compared to the ligand-bound forms, all systems exhibited comparable average RMSF profiles across the R12 domain, indicating no substantial changes in overall flexibility (see attached Fig). Based on these findings, we chose not to include the MD results in the main text, as they did not yield significant mechanistic insights beyond the crystallographic analysis.

Fig. Binding mode prediction through MD simulation. (A) RMSD and (B) RMSF curves from MD simulation are plotted.

Best regards,
 Zhiguang Yuchi

We thank the reviewers for the positive comment. I have reproduced your comments verbatim below together with our responses. For clarity, your criticisms and/or suggestions are coloured in dark red, while our response is shown in dark blue.

Reviewer #1

The revised ms by Hadiatullah et al examines the conformation of Repeat 12 (R12) domain of the ryanodine receptor (RyR), and binding affinity of the R12 domain with two inhibitors, dantrolene (DAN) and azumolene (AZU). The ms is generally well written, and the structural basis of inhibitor binding in RyR channels is likely to be of interest to a broad audience in structural biology, pharmacology, cell biology, and channel biophysics.

I am satisfied with the authors revisions and I have no further criticisms.

RESPONSE #: Thank you for your positive comments.

Reviewer #2

The authors have addressed all of my comments appropriately.

I only have a minor suggestion about a newly added paragraph for the authors to consider:

Changing a few word in this sentence "While the crystallographic studies of isolated R12 domain provide atomic-level detail, they lack the full protein context, potentially overlooking allosteric and membrane-associated effects." so that it becomes "While the crystallographic studies of the isolated R12 domain provide atomic-level detail, the full protein context is absent, potentially overlooking allosteric and membrane-associated effects." I think such a change can improve logical flow.

RESPONSE #: Thank you for your positive comments. We have changed the sentence according to your suggesrtion (Line 399-401)

Reviewer #3

The revised manuscript has addressed my concerns. Therefore, it is recommended that the manuscript be accepted for publication.

RESPONSE #: Thank you for your positive comments.

Best regards,
Zhiguang Yuchi